# Antispasmodic Effect of Asperidine B, a Pyrrolidine Derivative, through Inhibition of L-Type Ca^2+^ Channel in Rat Ileal Smooth Muscle

**DOI:** 10.3390/molecules26185492

**Published:** 2021-09-09

**Authors:** Acharaporn Duangjai, Vatcharin Rukachaisirikul, Yaowapa Sukpondma, Chutima Srimaroeng, Chatchai Muanprasat

**Affiliations:** 1Unit of Excellence in Research and Product Development of Coffee, Division of Physiology, School of Medical Sciences, University of Phayao, Mueang Phayao, Phayao 56000, Thailand; 2Center of Health Outcomes Research and Therapeutic Safety (Cohorts), School of Pharmaceutical Sciences, University of Phayao, Mueang Phayao, Phayao 56000, Thailand; 3Division of Physical Science and Center of Excellence for Innovation in Chemistry, Faculty of Science, Prince of Songkla University, Hat Yai, Songkhla 90110, Thailand; vatcharin.r@psu.ac.th (V.R.); yaowapa.suk@psu.ac.th (Y.S.); 4Department of Physiology, Faculty of Medicine, Chiang Mai University, Mueang Chiang Mai, Chiang Mai 50200, Thailand; chutima.srimaroeng@cmu.ac.th; 5Chakri Naruebodindra Medical Institute, Faculty of Medicine Ramathibodi Hospital, Mahidol University, Bangphli, Samutprakarn 10540, Thailand; chatchai.mua@mahidol.ac.th

**Keywords:** pyrrolidine derivative, relaxation, smooth muscle, ileum, calcium channel

## Abstract

Antispasmodic agents are used for modulating gastrointestinal motility. Several compounds isolated from terrestrial plants have antispasmodic properties. This study aimed to explore the inhibitory effect of the pyrrolidine derivative, asperidine B, isolated from the soil-derived fungus *Aspergillus sclerotiorum* PSU-RSPG178, on spasmodic activity. Isolated rat ileum was set up in an organ bath. The contractile responses of asperidine B (0.3 to 30 µM) on potassium chloride and acetylcholine-induced contractions were recorded. To investigate its antispasmodic mechanism, CaCl_2_, acetylcholine, Nω-nitro-l-arginine methyl ester (l-NAME), nifedipine, methylene blue and tetraethylammonium chloride (TEA) were tested in the absence or in the presence of asperidine B. Cumulative concentrations of asperidine B reduced the ileal contraction by ~37%. The calcium chloride and acetylcholine-induced ileal contraction was suppressed by asperidine B. The effects of asperidine B combined with nifedipine, atropine or TEA were similar to those treated with nifedipine, atropine or TEA, respectively. In contrast, in the presence of l-NAME and methylene blue, the antispasmodic effect of asperidine B was unaltered. These results suggest that the antispasmodic property of asperidine B is probably due to the blockage of the L-type Ca^2+^ channel and is associated with K^+^ channels and muscarinic receptor, possibly by affecting non-selective cation channels and/or releasing intracellular calcium.

## 1. Introduction

Antispasmodic compounds are widely used to control anxiety and musculoskeletal tension [1], and they are especially used for reducing intestinal motility in gastrointestinal smooth muscle spasms. Gastrointestinal motility disorders are described by abnormal intestinal contractions, including achalasia, non-achalasia esophageal motility disorders, dyspepsia, gastroparesis, chronic intestinal pseudo-obstruction, irritable bowel syndrome (IBS) and chronic constipation [2]. Abnormal intestinal contractions lead to impaired quality of life and high healthcare costs. Several drugs and alternative medicines have been proposed to manage intestinal motility disorders by intervening in the underlying pathophysiology of these disorders.

Gastrointestinal motility refers to smooth muscle contraction. The muscle contraction depends on the influx of calcium (Ca^2+^) through voltage-dependent L-type Ca^2+^ channels and releasing of Ca^2+^ from the intracellular store. The voltage-dependent Ca^2+^ channels are regulated by potassium (K^+^) and non-selective cation conductance in membrane potential and excitability [3]. Furthermore, the smooth muscle response depends on excitatory and inhibitory neurotransmitters, including acetylcholine, nitric oxide (NO) [3] and noradrenaline [4]. Acetylcholine interacts with muscarinic receptors and leads to elevation of intracellular calcium [5]. The NO/cyclic guanosine monophosphate (cGMP) pathway is involved in smooth muscle relaxation by activating soluble guanylyl cyclase (sGC) [6]. Similarly, the β-adrenoceptor triggers muscle relaxation [4].

Interestingly, most antispasmodic compounds are synthetic or semisynthetic compounds isolated from terrestrial plants [1]. Previous studies have demonstrated the antispasmodic effect of piperidine derivatives, phenacyl derivatives of 4-hydroxypiperidine and the piperidine analogue 1-(4′-fluorophenacyl)-4-hydroxy-4-phenyl-piperidinium chloride on blood pressure and smooth muscle contractions of isolated rabbit jejunum [7,8]. Tolperisone, a piperidine derivative, is suggested as a centrally acting muscle relaxant [9]. Ammonium pyrrolidine dithiocarbamate (PDTC) displayed a relaxation response on rat bladder smooth muscle induced by acetylcholine [10] and rat aortic smooth muscle contracted with phenylephrine [11]. Previously, we isolated a pyrrolidine derivative, asperidine B, from the soil-derived fungus *Aspergillus sclerotiorum* PSU-RSPG178 [12]. In this study, we tested the hypothesis that asperidine B will have an antispasmodic effect on isolated rat’s ileum motility. We also sought to investigate its mechanism by exploring its effect on the calcium influx, calcium channel, potassium channel, acetylcholine and NO/cGMP pathway.

## 2. Results

### 2.1. The Effect of Asperidine B on the Rat Ileal Contractile Response Induced by KCl

To investigate the relaxation effect of asperidine B on the ileal smooth muscle contraction induced by KCl, cumulative concentrations of asperidine B at 0.3 to 30 µM were added to the bath solutions. The results revealed that asperidine B inhibited the rat ileal contraction in a concentration-dependent manner and by ~37% at 30 µM and with IC_50_ ~58 µM, as shown in Figure 1. However, the difference in the inhibitory effect of asperidine B at 10 and 30 µM was not statistically significant. Therefore, the concentration at 10 µM was used for further investigation on the potential mechanism.

### 2.2. Mechanism of Asperidine B Action on Ileal Smooth Muscle Contraction

To identify the potential mechanism of asperidine B action related to the suppression of extracellular calcium influx, a cumulative concentration of CaCl_2_ (1–20 mM) was administrated to the organ bath containing a Ca^2+^-free solution in the absence or in the presence of asperidine B (10 µM). Adding cumulative calcium chloride to the bath led to an ileum contraction in a concentration-dependent manner. In contrast, the presence of asperidine B significantly reduced an inhibitory response of the contraction by ~14–17% compared to the control, as shown in Figure 2. These data revealed that the antispasmodic effect of asperidine B may involve the blockage of calcium influx. To further confirm that the antispasmodic effect of asperidine B was through the blockage of Ca^2+^ channels, nifedipine, which is an L-type Ca^2+^ channel blocker, was used in the absence or the presence of asperidine B. The results showed that neither nifedipine, asperidine B nor nifedipine combined with asperidine B exhibited an inhibitory effect on the ileum contraction induced by KCl, as shown in Figure 3. Nevertheless, there was no statistically significant difference between the groups, while nifedipine alone tended to decrease the contraction. These results suggested that the antispasmodic effect of asperidine B is probably due to the blockage of calcium channels and that different mechanisms may be involved.

To establish whether the antispasmodic effect of asperidine B was associated with the muscarinic receptor by involving a non-selective cation channel and/or releasing intracellular calcium stores, acetylcholine was tested in the absence or presence of asperidine B or atropine. Our results showed that contractile responses of various concentrations of acetylcholine were attenuated by asperidine B and atropine, as shown in Figure 4. However, there was no statistically significant difference in the contractile responses between asperidine B and atropine in induced contractions by acetylcholine. These data indicated that the antispasmodic property of asperidine B may be associated with modulating muscarinic receptors by affecting non-selective cation channels and/or the release of intracellular calcium.

To evaluate the involvement of K^+^ channels in the asperidine-B-mediated relaxant response, TEA was added to the bath. The presence of asperidine B, TEA and TEA combined with asperidine B revealed inhibitory activity on the ileum contraction when compared to KCl, as shown in Figure 5. Contractile responses of asperidine B were not changed by the non-selective K^+^ channel blocker TEA. However, TEA combined with asperidine B tended to inhibit the contraction more than TEA alone but did not reach a statistically significant difference between groups. For the ileal response, asperidine B alone suppressed the contraction better than KCl plus TEA, and TEA plus asperidine B. Additionally, there was no statistically significant difference between TEA and KCl plus TEA. These results indicated that antispasmodic responses of asperidine B may partly involve K^+^ channels.

To clarify that the relaxation effect of asperidine B is linked with NO production, the antispasmodic effect of asperidine B was investigated in the absence or presence of l-NAME, a NO synthase (NOS) inhibitor. The presence of asperidine B, l-NAME and l-NAME combined with asperidine B reduced the ileum contraction when compared to KCl, as shown in Figure 6a,b. There was no statistically significant difference between l-NAME and KCl plus l-NAME. Asperidine B alone and l-NAME plus asperidine B showed a significant inhibitory effect on the ileum contraction when compared to KCl plus l-NAME, but the difference was not statistically significant between groups. These results suggested that the relaxation effects of asperidine B may not involve NO production.

To further explore whether the asperidine B-induced ileal relaxation involved the NO–GC–cGMP (nitric oxide–guanyly cyclase–cyclic guanosine monophosphate) pathway, methylene blue, a GC inhibitor, was incubated in the bath. The presence of asperidine B, methylene blue and methylene blue plus asperidine B diminished the ileal contraction when compared to KCl, as shown in Figure 6c,d. Asperidine B alone and methylene blue plus asperidine B demonstrated a significant reduction in ileal contraction when compared to KCl plus methylene blue; however, there was no statistically significant difference between the groups. There was no statistically significant difference between methylene blue and KCl plus methylene blue. These results indicated that the relaxation effects of asperidine B may not involve the NO–GC–cGMP pathway.

## 3. Discussion

The contractility of gastrointestinal smooth muscle is an important regulator of gastrointestinal motility, in which abnormal contraction patterns can lead to dysmotility. High levels of K^+^ are known to induce membrane depolarization and activate smooth muscle contractions through the opening of voltage-dependent Ca^2+^ channels. Depolarization-dependent activation of voltage-gated Ca^2+^ channels can induce the release of Ca^2+^ from internal Ca^2+^ stores by opening ryanodine receptors without the influx of Ca^2+^ [13]. An increase in intracellular Ca^2+^ concentration activates Ca^2+^-calmodulin-dependent myosin light chain kinase (MLCK), resulting in a smooth muscle contraction [14]. Besides acetylcholine-induced smooth muscle contractions leading to gastrointestinal motility by activating muscarinic acetylcholine receptors (mAChRs), an increase in cytosolic Ca^2+^ concentrations and the stimulation of non-selective cation channels in the plasma membrane contribute to membrane depolarization and produce an influx of Ca^2+^ via voltage-gated Ca^2+^ channels [15,16].

In this study, the antispasmodic responses of asperidine B is given to pharmacological agents, for instance, nifedipine, atropine, tetraethylammonium and methylene blue, while acetylcholine, high K^+^ and CaCl_2_ were used to induce smooth muscle contraction. Our data indicated that asperidine B has antispasmodic effects on intestinal smooth muscle. A cumulative concentration of asperidine B suppressed the contraction induced by high K^+^. To establish the mechanism underlying the antispasmodic response, CaCl_2_ and nifedipine, an L-type Ca^2+^ channel blocker, were compared in the presence of asperidine B, which was shown to attenuate the contractile response induced by KCl. However, the relaxation response revealed a similar effect in the presence of KCl plus nifedipine, nifedipine combined with asperidine B or asperidine B alone. Meanwhile, nifedipine alone did not show response effect and difference with KCl plus nifedipine. This finding suggested that the antispamodic effect of asperidine B is possibly mediated through calcium channel blockage and involve other mechanisms. Moreover, our results displayed the antispasmodic effect of asperidine B by acetylcholine-induced contraction, a similar trend to that observed with atropine, a non-selective muscarinic receptor antagonist, blocking the contractile response. It has been proposed that 1-methyl-2-(2-methyl-1,3-dioxolan-4-yl)pyrrolidine and 1-methyl-2-(2-methyl-1,3-oxathiolan-5-yl)pyrrolidine derivatives can be binding and show selectivity at muscarinic receptors in Chinese hamster ovary (CHO) cells and in guinea pig and rabbit tissues [17]. Pyrrolidine derivatives are potent as inhibitors of cytosolic phospholipase A2α (cPLA2α), inhibiting eicosanoid production and blocking arachidonate release in human monocytic leukemia cells (THP-1 cells) stimulated with A23187 [18]. In intestinal smooth muscle, calcium mobilization contributes to the activation of cPLA2 and arachidonic acid (AA)-dependent stimulation of calcium influx [19]. Since acetylcholine-induced contractions depend on intracellular Ca^2+^ mobilization via the inositol 1,4,5-triphosphate (IP_3_) receptor on Ca^2+^ stores [20] and Ca^2+^ influx via L-type Ca^2+^ channels [21] through non-selective cation channel stimulation [15]. Our results might presume asperidine B to be an anticholinergic agent associated with the muscarinic receptor through calcium influx via L-type Ca^2+^ channels and/or the release of intracellular calcium. While potassium channels play a role in regulating the membrane potential and smooth muscle contractility [22,23]. High K^+^ induced contraction mediated by Ca^2+^ influx through voltage-operated Ca^2+^ channels [22], whereas TEA, a non-selective K^+^ channel blocker, blocks the Ca^2+^-activated K^+^ channels [24] and binds to the outer vestibule of the K^+^ channels [25], leading to a loss of hyperpolarization and relaxation. Interestingly, the antispasmodic effect of asperidine B was noticeably inhibited in the presence of TEA, indicating a mechanism involved in the K^+^ channels. As known neurotransmitters play a role on gastrointestinal smooth muscle contractility, including nonadrenergic noncholinergic (NANC) and NO. NO is produced by NOS and mediates smooth muscle relaxation via the activation of GC and the production of cGMP, leading to the stimulation of protein kinase G (PKG) and the inhibition of Ca^2+^ mobilization by inhibiting IP_3_ formation [26]. Additionally, asperidine B was insensitive to l-NAME, a NOS inhibitor and methylene blue, a GC inhibitor. This result indicates that asperidine B did not have an effect on NO production and was not mediated by the NO–GC–cGMP pathway. The results point out the antispasmodic potential of asperidine B through the blockage of calcium and potassium channels. Further investigations for electrophysiology and imaging experiments of Voltage-gated Ca^2+^ (CaV) channels subtypes, muscarinic signal pathways and molecular interaction studies of its interaction with binding sites are recommended.

## 4. Materials and Methods

### 4.1. Chemicals

NaCl, KCl, NaH_2_PO_4_, KH_2_PO_4_, MgCl_2_, CaCl_2_ and glucose were purchased from Ajax-Finechem (Australia). HEPES was obtained from PanReac AppliChem (Darmstadt, Germany). Acetylcholine (ACh), dimethyl sulfoxide (DMSO), ethylene glycoltetraacetic acid (EGTA), nifedipine, tetraethylammonium chloride (TEA), methylene blue and Nω-nitro-l-arginine methyl ester (l-NAME) were purchased from Sigma (St. Louis, MO, USA).

### 4.2. Pyrrolidine Derivative Asperidine B Material

Asperidine B (C_21_H_35_NO), a pyrrolidine derivative, was isolated from the soil-derived fungus *Aspergillus sclerotiorum* PSU-RSPG178, which was collected from the Plant Genetic Conservation Project under the Royal Initiation of Her Royal Highness Princess Maha Chakri Sirindhorn at Ratchaprapa Dam in Suratthani Province, Thailand. The specimen was deposited as BCC56851 at the BIOTEC Culture Collection, National Center for Genetic Engineering and Biotechnology (BIOTEC), Thailand, GenBank accession number KC478521, as described in previous reports [12,27,28].

### 4.3. Animals and Tissue Preparation

The animals (150–200 g) were purchase from Nomura Siam International, Pathumwan, Bangkok, Thailand. All experimental procedures were carried out in accordance with the ethical guidelines for animals. The experiment was approved by the Animal Ethics Committee of the University of Phayao, Phayao, Thailand (Approval No. 610204002). Male Wistar rats were maintained under a controlled environment at 25 °C with 12 h light/dark cycles at the Laboratory Animal Research Centre, University of Phayao. All rats were allowed free access to a standard diet and water. All animals were acclimatized for 1 week prior to the experiment.

Rats were fasted overnight and anesthetized with carbon dioxide (CO_2_) gas. The ileum was removed, cut into segments (1–1.5 cm) and then mounted in an organ bath containing Kreb’s solution (10 mM HEPES, 122 mM NaCl, 5 mM KCl, 0.5 mM NaH_2_PO_4_, 0.5 mM KH_2_PO_4_, 1 mM MgCl_2_, 1.8 mM CaCl_2_ and 11 mM glucose) at 37 °C and aerated with oxygen. The ileum tissue was fixed with a hook at the bottom of the bath, while another side was hanged with force transducer. The ileum was set up under 1 g tension and equilibrated in the bath for 60 min (washed every 15 min) before the start of the experiments. The contractile responses were recorded with the force transducer connected to an iWorx214 A/D converter and LabScribe2 program (iWorx Systems Inc., Dover, NH, USA).

### 4.4. Experimental Protocol

After the equilibration period, the tissues were induced to contract by KCl (80 mM), which was used as a maximum contractile control. After stable contractions by KCl, the responses of cumulative concentrations of asperidine B (0.3 to 30 µM) were tested. The contractile response of asperidine B was assessed relative to the maximum contraction (100%) produced by KCl and expressed as the percentage of the contraction. The maximum relaxation of asperidine B was used for further experiments.

To clarify that asperidine B acts through the blockade of extracellular calcium influx, cumulative concentrations of CaCl_2_ (1–20 mM) were added to a bath containing a Ca^2+^-free solution in the absence or in the presence of asperidine B (10 µM).

To investigate the involvement of asperidine B on an acetylcholine-induced contraction, acetylcholine (10^−8^–10^−5^ M) was added to the bath in the absence or in the presence of asperidine B (10 µM) or 100 nM atropine (a non-selective muscarinic antagonist).

To explore the potential mechanism of asperidine B against the contraction, 100 μM l-NAME (a nitric oxide synthase inhibitor), 1 µM nifedipine (a calcium channel blocker), 30 μM methylene blue (a guanylate cyclase inhibitor) and 5 mM TEA (a non-selective K^+^ channel blocker) were administered to the bath in either the presence or absence of asperidine B.

### 4.5. Statistical Analysis

All data are expressed as mean ± standard error of the mean (SEM) for each group of experiments (*n* = 5–9 for each set of experiments) and were analyzed by using a one-way analysis of variance (ANOVA) for repeated measures, and the significance of differences between groups were assessed by the Student’s *t*-test; *p*-values of less than 0.05 were considered significant.

## 5. Conclusions

These findings indicate that asperidine B possesses antispasmodic activities mediated predominantly through the blockade of L-type Ca^2+^ channel and K^+^ channels. In addition, its antispasmodic mechanism is associated with cholinergic receptors, probably by inhibiting Ca^2+^ influx and decreasing intracellular Ca^2+^. This study provides a rationale to use asperidine B in the development of antispasmodic agents and dietary supplements to protect against gastrointestinal dysmotility.

## Figures and Tables

**Figure 1 molecules-26-05492-f001:**
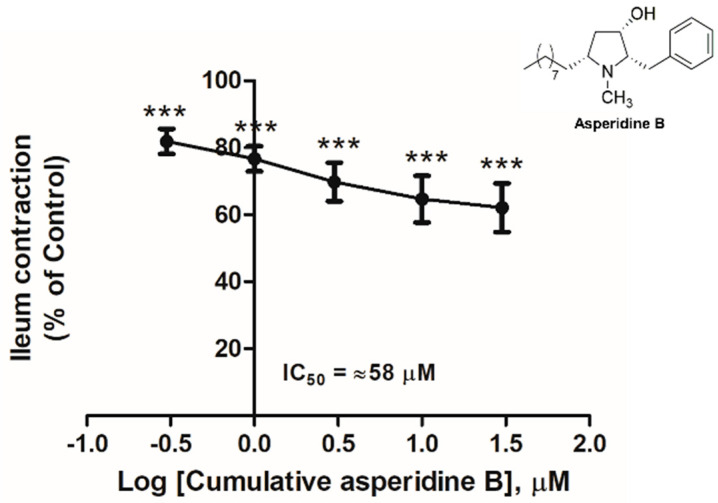
Antispasmodic effect of asperidine B on rat ileal contractions induced by KCl (80 mM, *n* = 6). *** *p* < 0.001 versus control.

**Figure 2 molecules-26-05492-f002:**
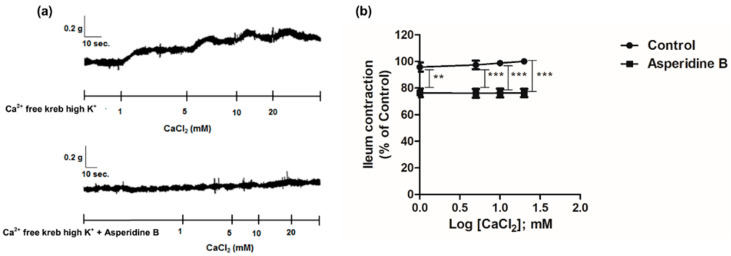
(**a**) Tracing and (**b**) activity of the contractile response of rat’s ileum smooth muscle to a cumulative concentration of CaCl_2_ in the absence or in presence of asperidine B (*n* = 6). *** *p* < 0.001; ** *p* < 0.005 versus control.

**Figure 3 molecules-26-05492-f003:**
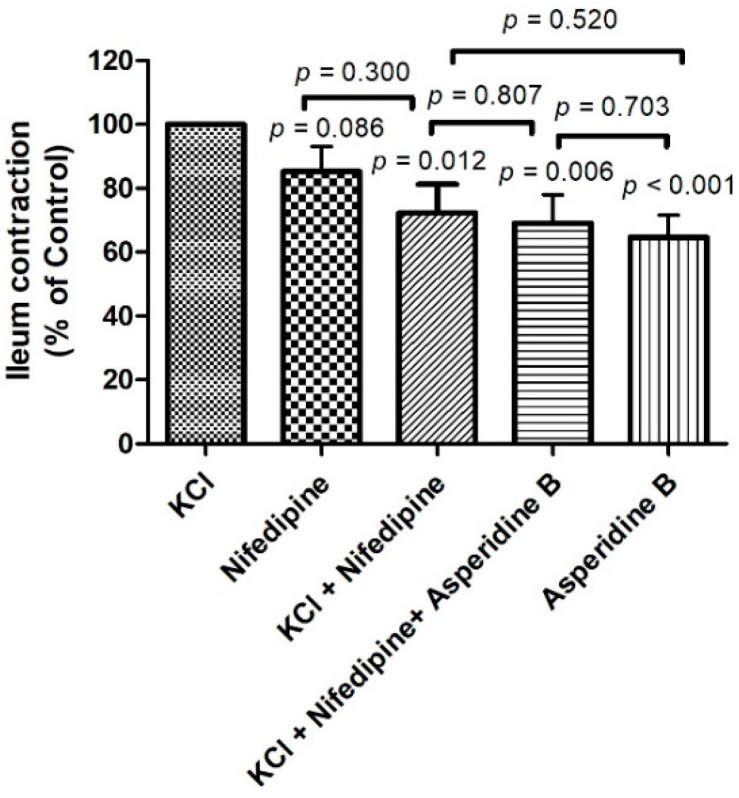
Effect of asperidine B on rat’s ileum contractions induced by KCl in the absence or presence of 1 µM nifedipine (*n* = 6). The differences of the tested compounds, except specific comparator, indicated that it compared with KCl.

**Figure 4 molecules-26-05492-f004:**
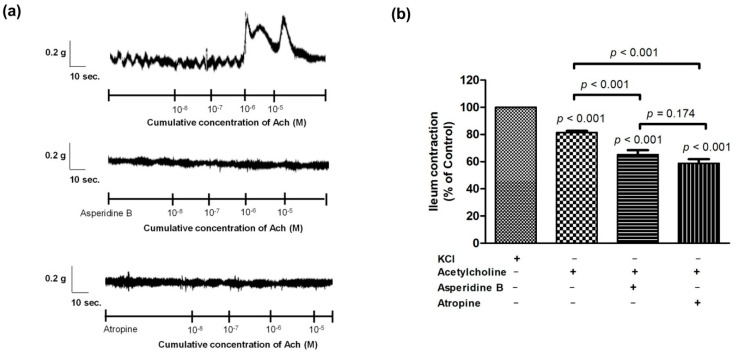
(**a**) Representative tracing and (**b**) activity of the contractile response of acetylcholine (10^−8^–10^−5^ M) in the absence or presence of asperidine B or atropine. Effect of acetylcholine (10^−5^ M)-induced contractions in the absence or presence of asperidine B or atropine (*n* = 6). The differences of the tested compounds, except specific comparator, indicated that it compared with KCl.

**Figure 5 molecules-26-05492-f005:**
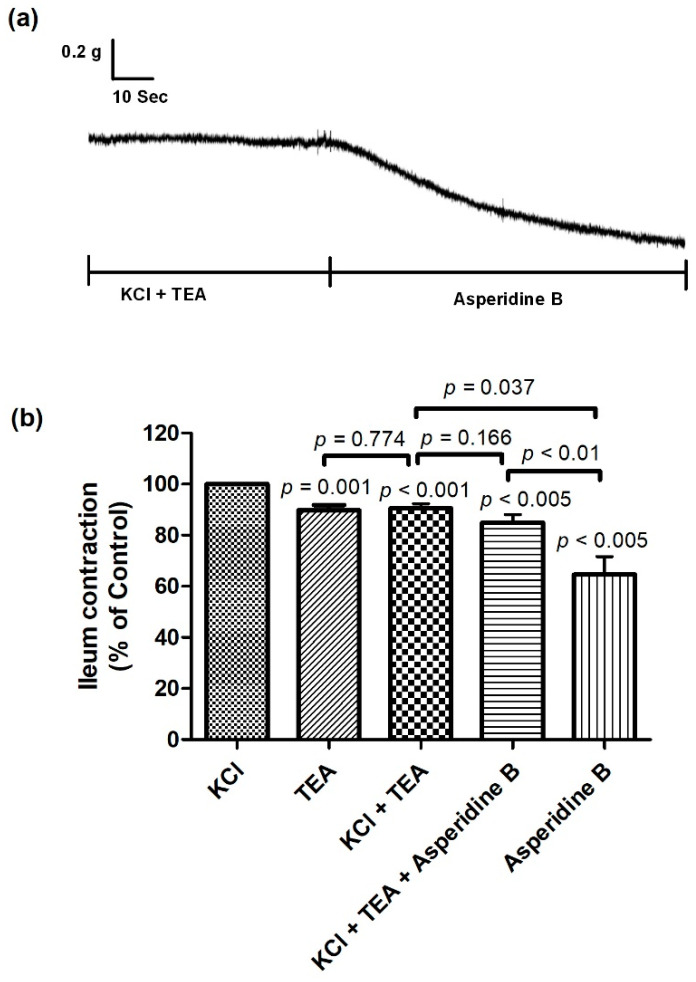
(**a**) Representative tracing and (**b**) effect of asperidine B on rat’s ileum contractions induced by KCl in the absence or presence of TEA (*n* = 5–6). The differences of the tested compounds, except specific comparator, indicated that it compared with KCl.

**Figure 6 molecules-26-05492-f006:**
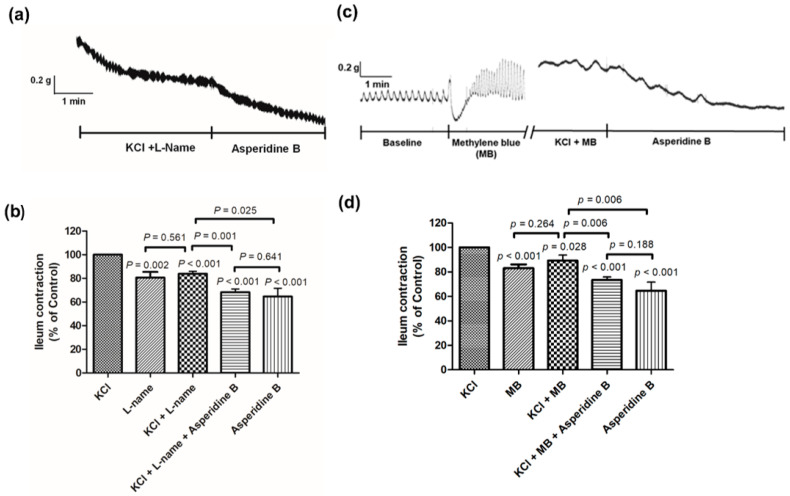
Effect of asperidine B on rat ileal contractions induced by KCl in the absence or presence of (**a**,**b**) l-NAME (*n* = 6) or (**c**,**d**) methylene blue (*n* = 6–9). The differences of the tested compounds except specific comparator indicated that it compared with KCl.

## Data Availability

The data presented in this study are available upon request from the corresponding author.

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
