# Peer review of "Antispasmodic Effect of Asperidine B, a Pyrrolidine Derivative, through Inhibition of L-Type Ca2+ Channel in Rat Ileal Smooth Muscle"

_molecules, 2021, doi:10.3390/molecules26185492_

Round 1

Reviewer 1 Report

General points:

.) According to the description in the “Experimental Protocols”, the authors used “whole” - ileal preparations and not isolated smooth muscles. A major concern in the interpretation of the data is, whether the observed effects are indeed caused by the action on smooth muscles directly, or if it might be mediated via influencing other cells, foremost neuron, present in the preparation. At least from vascular preparations it is known that remnants of sympathetic nerve endings are viable in such preparations for periods of more than one hour and capable of still releasing norepinephrine upon electrical stimulations. This is a severe problem for inferring the mechanism of action of asperidine B and should be addressed, at least in the discussion section.

.) The authors should present sample traces for all the experiments, so the reader could better appreciate the findings.

.) There are several typos and grammar errors throughout the manuscript that should be fixed.

Specific points:

Experimental protocols:

From the experimental protocol section it is not entirely clear how the contraction data has been analyzed. Is the tension data presented the “raw” tension measured (i.e. the 1 g basal tension +/- any changes) or the change in tension alone? If the “raw” tension is reported throughout the manuscript, I strongly recommend to change the analysis, as this is quite misleading and can cause strange results (see points below).

Section 2.1 / Figure 1:

Please fit the datapoints using a concentration response relationship and report IC50 values including their 95% confidence intervals. I also recommend to present it in a semilogarithmic fashion so the concentration dependence (including Hill coefficients) is easier to interpret.

Section 2.2 / Figure 2:

There is a mismatch between the sample trace in the upper panel of Figure 2 (a) and the corresponding trace in (b). While in (a) there is a clear effect of increasing CaCl2 concentrations they are barely visible in (b). I assume this is caused by the way data has been analyzed, as I assume that the authors report the “raw” tension data (see above).

Section 2.2 / Figure 3:

It is more than surprising, that Nifedipine was only able to block about 30 % of the KCl induced contraction. KCl should induce mainly via Cav1.x channel activation and thus should be blocked to large amount by the addition of a dihydropyridine (as it has been reported numerous times in literature). Again this discrepancy could be caused by the way data are presented.

Section 2.2 / Involvement of Ca2+ channels:

I am not fully convinced by the authors conclusions concerning the involvement of Cav1.x. The presented data is at best a hint towards the involvement of Cav1.x channels. I would thus recommend to test asperidine B on heterologously expressed Cav.1.x channels in electrophysiological experiments. Using this method concentration dependencies and modes of action can be studied in a much cleaner fashion. I understand that this method might not be readily available to the authors, but there are many labs that perform such experiments on a regular basis and are willing to collaborate.

Section 2.2 / Figure 4: Please show the concentration response relationship for ACh. The sample trace in Figure 4 (a) upper panel, besides being very unstable also before the addition of ACh, shows no response before 1 µM  and then suddenly saturates in its effect. This is unsettling. I also disagree with the conclusions in this case. If the conclusions from figure 2 are correct, the effect on Cav1.x channels will also be present in this experiment. So, to correctly demonstrate the effects on muscarinic receptors, the experiment would have needed to be performed in the presence of nifedipine to exclude Cav1.x effects. As an alternative, the authors could perform experiments with heterologously expressed muscarinic receptors (e.g. measure IP3 production or Ca2+ imaging) to study asperidine’s action on muscarinic receptors.

Section 2.2 / Figure 5 & 6: Similar to Figure 4 the data interpretation is complicated by the possible action of asperidine on calcium channels.

Author Response

We would like to thank reviewers for careful and thorough review of this manuscript. We have revised our manuscript in response to your suggestions and the changes have made as highlighted in “red font”. We hope that this improved manuscript is acceptable for publication in Molecules.The answer to their specific comments/suggestions are as follows.

Reviewer 2 Report

Duangjai et al presented a set of very interesting data studying the potential mechanisms of the antispasmodic effect of asperidine B on spasmodic activity. They used an array of pharmacological blockers to show that the antispasmodic effect of asperidine B is mediated through L-type calcium channels, K+ channels and muscarinic receptor.

The findings are of interest to the gastrointestinal and ion channel research communities in general. The experiments are well planned out and executed, and the manuscript is very well written. It will be a good fit for the journal of Molecules. However, I would like to see the following points being addressed before its publication at Molecules:

Figure 1, please re-plot x-axis with log scale instead of linear scale. If possible, please fit the data and report the IC50 of asperidine B under this condition.

Subtitle 2.2: should be “mechanisms of asperidine B action …”

Figure 2 (a), for the asperidine B condition, the authors should also show the baseline before asperidine B is added (if the authors recorded the contraction)

For Figures 3-6, the key message lies in the comparison between the 2nd vs 3rd bars, especially when there is no statistical significance (thus the occlusion effect by the respective blockers). Therefore, please label clearly ns if there lacks a significant difference between 2nd and 3rd bars.

Author Response

(The authors gave the same response as above.)
